# Qualitative evaluation of an edutainment intervention to prevent age-disparate transactional sex in Tanzania: Changes in educational aspirations and gender equitable attitudes towards work

**Marjorie Pichon**[1][☯]*, **Ana Maria Buller**[1][☯], **Veronicah Gimunta**[2], **Oscar Rutenge**[3], **Yandé Thiaw**[1], **Revocatus Sono**[2], **Lottie Howard-Merrill**[4]

1 Gender Violence & Health Centre, Department of Global Health and Development, London School of Hygiene & Tropical Medicine, London, United Kingdom, 2 Amani Girls Organization, Mwanza, United Republic of Tanzania, 3 Tanganyika Christian Refugee Service, Dar es Salaam, United Republic of Tanzania, 4 Department of Education, Practice and Society, Institute of Education, University College London, London, United Kingdom

☯ These authors contributed equally to this work.

* Marjorie.pichon@lshtm.ac.uk

**Data Availability Statement:** The research is based upon an analysis of in-depth interviews

## Abstract

Age-disparate transactional sex is a major contributor to the disproportionate rates of HIV experienced by adolescent girls in sub-Saharan Africa, and a key driver of unintended adolescent pregnancy. This paper comprises one element of the impact evaluation of the *Learning Initiative on Norms, Exploitation and Abuse (LINEA)* radio drama intervention to prevent age-disparate transactional sex. It provides new insights into the radio drama's influence on distal drivers of age-disparate transactional sex identified in formative research: girls' own educational aspirations, and gendered attitudes towards work. The intervention, which targeted adolescent girls and their caregivers in the Shinyanga Region of Tanzania, uses an edutainment approach to prevent transactional sex between girls aged 12–16 years and men at least 5–10 years older. We distributed the 39-episode radio drama on USB flash drives to 331 households and conducted longitudinal in-depth interviews with 59 participants. We conducted a thematic analysis of endline (December 2021) transcripts from 23 girls, 18 women caregivers, and 18 men caregivers of girls (n = 59), and midline (November 2021) transcripts from a sub-sample of these participants: 16 girls, 16 women and 13 men (n = 45). Findings suggest the radio drama created an enabling environment for preventing age-disparate transactional sex by increasing girls' motivation to focus on their studies and remain in school. There was also strong evidence of increased gender-equitable attitudes about work among girls and women and men caregivers. These supported women joining the workforce in positions traditionally reserved for men and challenging the male provider role. Our findings suggest that the *LINEA* radio drama can supplement interventions that address structural drivers of age-disparate transactional sex. The radio drama may also have impacts beyond preventing age-disparate transactional sex, such as reducing girls'

conducted with a small sample of adolescent girls and caregivers. These transcripts cover topics that are considered sensitive by participants and contain context-specific information that would enable them to be identified if transcripts were made available in their entirety. To safeguard the confidentiality and welfare of the individuals interviewed, we are therefore only able to share excerpts of anonymised transcripts that underpin the conclusions drawn in our manuscript. These excerpts may be requested for use in ethically approved research via the LSHTM Data Compass repository at https://doi.org/10.17037/DATA.00003750 or by emailing researchdatamanagement@lshtm.ac.uk.

**Funding:** This research was funded by the OAK Foundation (Grant No. OFIL-20-236), Wellspring Foundation (Grant No. 13343) and FELM (Finnish Evangelical Lutheran Mission) (Grant No. TZ710). The funders had no role in study design, data collection and analysis, decision to publish, or preparation of the manuscript.

**Competing interests:** The authors declare that no competing interests exist.

HIV morbidity and mortality, and challenging attitudes that promote sexual and gender-based violence to foster more gender-equitable communities across Tanzania.

## 1 Introduction

Adolescent girls and young women in sub-Saharan Africa experience disproportionate rates of morbidity and mortality, in large part driven by HIV [1–4]. Age-disparate transactional sex is a major contributor to this high HIV prevalence [5, 6], as it has been found to co-occur with risky sexual behaviours such as unprotected sex and mixing sex and alcohol [7]. Age-disparate transactional sex is defined as occurring when an adolescent girl and man at least 5–10 years older have sex with the implicit understanding that material support or other benefit will be given in return [8]. These relationships are also associated with other harmful sexual and reproductive health outcomes including sexual violence, unplanned and early pregnancy and unsafe abortion [7]. Adolescent pregnancy, in turn, has been associated with higher rates of maternal and perinatal mortality, in addition to other negative outcomes for babies and young mothers [9].

The link between age-disparate transactional sex and educational attainment, defined as "the highest level of education completed by a person" [10], although established in the literature has received less attention. We know that for adolescent girls in sub-Saharan Africa there is a complex and important relationship between educational attainment, money and HIV, and that age-disparate transactional sex relationships seem to play a key role in consolidating these relationships [11–13], but the nuances of these associations are less clear. Remaining in school for instance, has been found to be a key protective factor for HIV [14–17], with in-school girls less likely to engage in sex, including age-disparate sex, than their out-of-school peers [18]. Education, however, is not in itself a 'silver bullet' for girls' sexual and reproductive health outcomes. A recent systematic review and meta-analysis examining the effects of education interventions on sexual and reproductive health in low- and middle-income countries found inconclusive results, and the authors highlighted the need to better understand in which circumstances and contexts schooling is beneficial [19]. For instance, there is evidence from a four-country study in sub-Saharan Africa that some adolescent girls may engage in age-disparate transactional sex to pay for school fees, school materials, and transport to school, and once they have engaged in premarital sex, they are at a higher risk of leaving school due to pregnancy or early/forced marriage [18]. In Tanzania, there is also evidence that inequitable gender norms are reproduced within schools, such as between students and teachers, and reports indicate that many in-school adolescent girls face sexual harassment and abuse from male peers and teachers [20, 21].

Widespread investment in girls' education following the introduction of the Millennium Development Goals and the Sustainable Development Goals, for example through the introduction of fee-free primary school across sub-Saharan Africa, has improved adolescent girls' access to primary education. Gender inequitable norms, however, have played a large role in upholding persisting attainment disparities between adolescent girls and boys in secondary and tertiary education [16, 18, 22]. Educational attainment for Tanzanian adolescent girls is particularly low; they have one of the lowest transition rates from primary to secondary school in sub-Saharan Africa [23], and of those adolescent girls who do enrol in secondary education, only 27% complete it [24].

In 2017 a controversial and highly contested new education policy in Tanzania led to the introduction of mandatory pregnancy testing and expulsion of pregnant girls [25]. This policy was challenged in 2021 when six girls who were expelled from school because they were pregnant filed a lawsuit against the Tanzanian Government, leading the Ministry of Education to ban pregnancy tests in schools, and require schools to allow pregnant and married adolescent girls and young women to continue with their education [26]. The enforcement and implementation of these new policies on schools, and their long-term impact have yet to be seen; including on the educational attainment of adolescent girls, as well as their HIV status and overall sexual and reproductive health. However, they show a step in the right direction to alleviate the gendered educational gap in Tanzania.

Amid the current state of flux in national gendered educational policies, there exists a critical need for research that evaluates interventions designed to address and rectify inequitable gender norms within the realm of education [27, 28]. A recent literature review identified fourteen interventions in sub-Saharan Africa (ranging from material support to community-based interventions) that demonstrated promise in increasing adolescent girl's educational attainment [29]. The primary aim of most of these interventions was HIV prevention, and educational outcomes were secondary. Findings suggested that material support interventions were most successful, while less is known about interventions aimed at changing attitudes and behaviours [29]. Importantly, none of the interventions included in the review aimed to change gender norms, and none were conducted in Tanzania [29]. While some promising interventions are currently being evaluated in the country, most notably the USAID funded *Waache Wasome* or "Let them Learn" project [30], a gap persists in gender norms interventions promoting adolescent girl's educational attainment.

The current study evaluates the *Learning Initiative on Norms*, *Exploitation and Abuse (LINEA)* radio drama intervention, which is the first, to our knowledge, with the primary goal of preventing age-disparate transactional sex. Formative research informing intervention development suggested that risk of engaging in age-disparate transactional sex relationships related to girls' motivations and beliefs about education. Wamoyi and colleagues found that age-disparate transactional sex caused interesting tensions between Tanzanian adolescent girl's short-term and long-term aspirations [27], whereby adolescent girls' participation in age-disparate transactional sex tended to reflect their short-term aspirations, which included dressing well and appearing attractive to men and their peers [27, 28, 31]. This was in direct contradiction with their long-term educational goals, which were linked to wealth, respect, freedom and independence, and were conversely put at risk by the sexual and reproductive health consequences of engaging in age-disparate transactional sex at an early age, in particular due to unplanned pregnancy [27].

In developing the *LINEA* intervention, we chose to use an entertainment-education or 'edutainment' approach, mixing important health messages with entertaining content in the form of a radio drama. We utilised this approach as it has proven successful in promoting a wide range of health behaviours across large and varied audiences [32], including sexual and reproductive health in the United States [33] and gender-based violence in rural Senegal [34]. The edutainment approach is evidence-based and grounded in social science theory, in particular Albert Bandura's social cognitive theory, which suggests that audience members are more likely to engage in new behaviours, or modify existing behaviours in line with those performed by characters if they like the characters or identify with them [35]. Edutainment is considered to be successful because it role models positive behaviours and the beneficial consequences that result from them, rather than lecturing against negative behaviours, as is common in more traditional, didactic programming [32].

In this paper we explore the potential of the *LINEA* radio drama intervention to help adolescent girls refocus on and prioritise their long-term aspirations of building a better future. This reinforces the more direct intervention messaging about age-disparate transactional sex, as reported in our companion paper [36], by highlighting that any benefits from age-disparate transactional sex are short-term, and that age-disparate transactional sex can lead to multiple long- and short-term harms to girls' education and aspirations, in addition to health-related problems. Findings are from an exploratory, mixed-methods evaluation of a condensed, radio drama-only version of the full *LINEA* intervention (comprised of curriculum and radio drama components). Data was collected in Shinyanga region, Tanzania from September-December 2021, and found encouraging shifts in knowledge, behaviours, and attitudes, beliefs and social norms linked directly to age-disparate transactional sex [36]. In the current paper we report additional results from the qualitative data on indications of desirable change that while not linked directly to age-disparate transactional sex, were still in line with the goal of promoting adolescent girl's sexual and reproductive health: 1) adolescent girl's focus on long-term educational goals, and 2) shifts towards more gender equitable attitudes about work. A forthcoming process evaluation paper is also planned, which will include evidence of unintended consequences from the radio drama.

This study was a collaboration between the Tanganyika Christian Refugee Service (TCRS) and Amani Girls Organization (AGO) in Tanzania, and the London School of Hygiene & Tropical Medicine (LSHTM) in the UK. The ultimate aim of this study was to inform a cluster randomised controlled trial (cRCT) of the *LINEA* intervention planned for 2023–2026 in Mwanza, Tanzania.

## 2 Methods

### 2.1 Ethics statement

This study has ethical approval from the National Institute for Medical Research (NIMR) in Tanzania (Ref: NIMR/HQ/R.8a/Vol.IX/3698) and LSHTM (ref: 22863–1). We followed rigorous protocols around ethics and confidentiality when conducting research on sensitive issues and with children, including ensuring ongoing informed consent and respecting the dignity of all participants. AGO and TCRS collected written informed consent from all participants and parents or guardians of all participants under 18 years old in this study using: 1) a participant information sheet, 2) a comprehension questionnaire and 3) an informed consent form. We also collected written informed assent from all participants under 18 years old—whose parents or guardians had provided written informed consent as described above—using 4) an informed assent form.

Crucially, our safeguarding referral protocols were developed in partnership with AGO and TCRS, who have extensive experience working in this (geographical and programmatic) area, as well as an established referral network. Our protocols also met LSHTM research integrity guidelines. Participants who required psychosocial support were provided referrals, and their cases were followed up by AGO staff. For further information about the ethics frameworks and guidelines drawn upon in this study see our past publications [37, 38].

### 2.2 Intervention and study design

Interventions aimed at preventing HIV, or negative sexual and broader reproductive health outcomes, have found mixed impacts on the outcomes identified in this study: increased focus on educational aspirations and shifts towards more gender equitable attitudes about work. Some of the most promising are the multi-component *DREAMS* intervention and *Stepping Stones*. *DREAMS* was found to be effective at moderately increasing adolescent girl and young

women's educational aspirations, as well as school retention and reenrolment in Nairobi, Kenya [39], but did not find any impacts on gender equitable norms [40]. The *Stepping Stones* study in Karnataka, India found that intervention participants had more gender equitable attitudes around girl's education compared to non-intervention participants [41], but did not measure school retention or reenrolment as behavioural outcomes. These findings reinforce emerging evidence that individual attitudes are easier to shift than social norms [42] and behaviours [43].

The *LINEA* intervention was developed to work beyond changes in individual knowledge and attitudes to impact long-term norm and behaviour change at the community level. The *LINEA* intervention was iteratively co-developed by LSHTM, AGO and Media for Development International in Tanzania [44] and comprises of two components: 1) a radio drama, and 2) two curricula, one targeting adolescent girls and the other men. The current study evaluated the radio drama component only, which includes thirty-nine 15–20 minutes episodes, and follows the central character Amali, a 13-year-old secondary school girl. Amali is pursued by Tuma, a 23-year-old *bodaboda* (motorbike taxi) driver who pressures her to accept free lifts to school, and subsequently demands sex in return. Throughout the radio drama both characters are given helpful and harmful advice by others in their community, and listeners follow as they decide on the best course of action.

For this evaluation the *LINEA* radio drama was distributed on USB flash drives to 331 TCRS beneficiary households across 24 villages in the Kishapu district of Shinyanga. These beneficiary households were selected on the basis of having a family member living with a disability and being the poorest in the community. Participants were recruited from 21–27 July 2021. Participating households were told to listen to five episodes per week on solar powered radios provided by TCRS. They were, however, free to listen to each episode as many times as they wanted to, and with whomever they wished. TCRS also facilitated household level discussion sessions about radio drama content with 60 randomly selected households to encourage participation. TCRS facilitators were volunteer members of the intervention community and were extensively trained by AGO.

To qualitatively evaluate indications of change on adolescent girls and their caregivers, 81 households with adolescent girls aged 12–16 years were randomly assigned to take part in longitudinal baseline in-depth interviews (IDIs) in September 2021 and endline IDIs in December 2021 immediately post-intervention. Households were randomly assigned to data collection with an adolescent girl (n = 27), or their female (n = 27) or male caregiver (n = 27). Additional midline IDIs were conducted in November 2021 with the 60 randomly selected participants who partook in household discussion sessions. Participants with physical disabilities were included in data collection, while those with cognitive disabilities were excluded. In total, implementation occurred over a seven-week period. For all waves of data collection Kiswahili-speaking researchers from AGO conducted sex-matched interviews, following a 10-day training and 5-day refresher training on interview techniques, ethics, safeguarding, data handing and processing and safety protocols. See [36] for more information on ethical considerations, study participants, sampling, the intervention, the study design and data collection procedures.

## 2.3 Study setting

Located in East Africa, Tanzania encompasses over 120 ethnic groups united under two official languages, Kiswahili and English. While roughly one-third of the population continues to practice traditional religions, the remainder are evenly split between Islam and Christianity [45]. Tanzania has the third fastest growing economy in Africa [46], with over half of its

population under the age of 18 years [47]. Approximately one-quarter of Tanzanian women will experience sexual violence before the age of 18 [48], and in a National cross-sectional survey conducted with approximately 15,000 adolescent girls and young women aged 15–24 year-old, there was a reported 43% lifetime prevalence of transactional sex [6, 49]. Our study was conducted in Kishapu district of Shinyanga region, located in North-western Tanzania, south of Lake Victoria, where the largest ethnic group is Sukuma [45]. Shinyanga has particularly high rates of gender-based violence [50], including the highest in the country for child marriage [51], and strong gender inequitable norms, particularly linked to sexual violence [52].

## 2.4 Data analysis

For this study we conducted a mixed inductive and deductive thematic analysis of the midline and endline IDIs using NVivo 12 [53]. While the midline IDIs were conducted with a sub-sample of the endline participants, we did not analyse this data longitudinally as we did not find evidence of any meaningful differences between the two timepoints. Moreover, given that they were conducted only three and a half weeks apart any small differences that were observed were likely due to the same questions being asked differently. Our analysis specially focused on participant's responses to questions about their experiences with and perceptions of the *LINEA* radio drama, and whether it impacted their future aspirations. Thus, we did not include transcripts from baseline IDIs, which were conducted before participants were given the radio drama.

We began by reading the transcripts repeatedly to become familiar with their contents. As we generated key themes from the data, we recorded them in a separate document to create the initial coding framework. Three researchers then used the preliminary coding framework to code the same three transcripts, one from an adolescent girl, one from a woman caregiver and one from a man caregiver. The researchers met to discuss challenges with and discrepancies between their coding, using this conversation to refine the coding framework. Next, we coded the remaining transcripts, and dual coded 18 of the richest ones in their entirety. We discussed the key thematic findings together and took note of the most poignant quotes. Lastly, we shared the findings with the wider research team, getting feedback from the Principal Investigator and local researchers to add contextual information to our results.

## 3 Results

We begin this section by describing the study participants. We then present evidence of additional results [to complement our primary findings [36]] from the evaluation of the *LINEA* radio drama that related to underlying attitudes, beliefs and social norms that helped create an enabling environment for preventing age-disparate transactional sex: 1) adolescent girls focusing on long-term educational goals, and 2) increased gender equitable attitudes about work among adolescent girls and women and men caregivers. While the first theme was specific to adolescent girls, the second theme arose among adolescent girls, women caregivers and men caregivers, however, we did not find evidence of any notable differences between them. Thus, findings from all three groups are reported together.

### 3.1 Study participants

Of the 81 participants recruited, 59 (73%) partook in endline IDIs. Drop-out rates were highest among adult participants, the most common reasons being busy schedules and work-related travel. We analysed 104 transcripts collected at midline and endline from these 59 participants. Endline transcripts were available from 23 adolescent girls, 18 women and 18 men, and midline transcripts were available from a sub-sample of these participants: 16 adolescent girls, 16

**Table 1. Number of transcripts analysed for this study from midline and endline in-depth interviews (IDIs) with 81 adolescent girls, women caregivers and men caregivers.**

|  | Adolescent girls | Women caregivers | Men caregivers | All |
|---|---|---|---|---|
| Midline IDI transcripts | 16 | 16 | 13 | 45 |
| Endline IDI transcripts | 23 | 18 | 18 | 59 |
| Total | 39 | 34 | 31 | 104 |

women and 13 men (Table 1). The majority of adolescent girl participants were 15 years old (n = 10), while women's ages ranged from 31-years to 77-years, and men's ages ranged from 25-years to 76-years.

## 3.2 Adolescent girls focusing on long-term educational goals after listening to the radio drama

Participants included in this study were the most deprived in their community and had a family member living with a disability. Therefore, the radio drama messages of focusing on long-term goals to build a better future, rather than risking long-term harm to meet short-term needs through age-disparate transactional sex, were particularly pertinent. We found evidence of change from transcripts with adolescent girls with regards to focusing on their long-term educational goals in three thematic areas: 1) Motivation to focus on studies, 2) Ability to move past life challenges, and 3) Avoiding sexual and reproductive health risks. Despite being asked similar questions, these themes did not arise in interviews with women and men caregivers.

**3.2.1 Motivation to focus on studies.** Encouragingly, one of the biggest reported impacts of the intervention (including when compared to findings directly liked to age-disparate transactional sex), and described by most adolescent girls, was increased motivation to focus on their studies:

> *I was lazy when it came to studying, so when I listened to the drama, when Tana [adolescent girl character] was insisting her fellow students study hard, then I [. . .] thought if I study hard, I can change my family life and I can secure a job.* (Adolescent girl aged 15-years, endline IDI 4)

> *I changed [after listening to the radio drama], now I want to study hard so I can reach my dreams.* (Adolescent girl aged 15-years, endline IDI 123)

In these quotes we see how the messages promoting the importance of education influenced the adolescent girl listeners. For example, in the radio drama the central character Amali's best friend is Tana, who is also her most ambitious schoolmate. Tana expresses her gender equitable opinions boldly and encourages her friends to continue studying, even when they face setbacks, so they can achieve their goals. There was evidence that Tana's character served as a role model for some adolescent girls to emulate, encouraging their classmates to also focus on their education:

> *Tana [. . .] [encouraged me to] study hard and advise my fellow student to do the same.* (Adolescent girl aged 13-years, endline IDI 20)

One participant, however, highlighted the challenges adolescent girls face in school because of sexual harassment by teachers, which was also a storyline in the radio drama. When asked her favourite scene, she responded:

*When teacher Pindi was arrested because he was manipulating people [students, into age-disparate transactional sex]. The [girl] students, they are supposed to be students and not [. . .] sex objects, because they are there [at school] to learn.* (Adolescent girl aged 15-years, endline, IDI 4)

**3.2.2 Ability to move past life challenges.** For a few adolescent girls the radio drama also increased their perceived ability to use education as a route to move past life's challenges. For example, when asked what she learned from the radio drama an adolescent girl responded:

*[I learned from the radio drama] not to lose hope and to make an effort on studying. . . When she [Amali] failed she didn't lose hope, she made more [of an] effort, and I will do the same.* (Adolescent girl aged 15-years endline, IDI 7)

The radio drama included a range of adolescent girl characters, who seemed to provide adolescent girl listeners with different sources of inspiration and modelled different ways to overcome life's challenges. Another adolescent girl character was Nyota, Amali's older sister who had engaged in age-disparate transactional sex with teacher Pindi, become pregnant, and was expelled from school before becoming a motorcycle taxi driver herself. When asked if she was inspired by any of the characters in the radio drama, another adolescent girl responded:

*[I was inspired by] Tana, she was educating her fellow students to study hard, she told Amali not to give up. [. . .] [I was also inspired by] Nyota, when she got pregnant, she didn't give up. That means when [you are] facing problems, you don't have to give up.* (Adolescent girl aged 15-years, endline IDI 4)

For an out-of-school adolescent girl, Nyota's storyline highlighted opportunities to return to education and work post pregnancy, and provided hope that there were still ways she could achieve professional success:

*When the girl [Nyota] was expelled from school because of pregnancy, still later she can fulfil her goals. . . I gave up on studying but now I want to know how to do tailoring.* (Adolescent girl aged 13-years, endline IDI 34)

Although this finding was only seen in one participant, it suggests that the radio drama may hold promise in motivating out-of-school adolescent girls as well as those in-school to reach their goals, and challenge prevailing narratives about pregnant schoolgirls lives being 'destroyed'.

**3.2.3 Avoiding sexual and reproductive health risks.** Notably, a few adolescent girls also reported learning about sexual and reproductive health risks through the radio drama, and how these could impact their ability to take part in education and reach their long-term goals. There was evidence that this encouraged some adolescent girls to wait to have sex, in particular to avoid becoming pregnant:

*Because I saw Nyota when she got pregnant, she was expelled from school. . . [so I learned] not to engage in sex. . . so I can reach my dream.* (Adolescent girl aged 15-years, endline IDI 29)

Similarly, when asked what she learned during the household discussion sessions about the radio drama, an adolescent girl reported learning that sex should be avoided to prevent long-term negative consequences:

*[The important thing I learned was] to stop sexual activities [. . .] [because they will cause me] to get pregnant while I am still young.* (Adolescent girl aged 14-years, endline IDI 3)

In these quotes we see how increases in knowledge attributed to the radio drama, as well as the modelling of aspirational behaviours, inspiration and encouragement could help prevent future risky sexual behaviours directly. In line with edutainment theory, we also see that this was because of the impact listeners heard that risky sexual behaviours could have on adolescent girl character's educational goals.

### 3.3 Adolescent girls and caregivers increased gender equitable attitudes about work after listening to the radio drama

We found evidence from adolescent girls, and women and men caregivers that the radio drama led them to adopt more gender equitable attitudes about work, particularly in relation to: 1) Women working in traditionally male-dominated professions, as well as change in 2) Women's motivation to engage in income generating activities. The evidence for the second theme, however, came solely from interviews with women caregivers.

**3.3.1 Women working in traditionally male-dominated professions.** These accounts arose primarily in relation to Nyota, an adolescent girl character who becomes a *bodaboda* (motorcycle taxi) driver, an occupation typically reserved for men.

*When Nyota went to be a bodaboda mechanic, at first, I was wondering how can she do it? But later I understood that there is no work for men only, us girls we can do any work. Just in the community when you do men's work, they [the community] will wonder.* (Adolescent girl aged 15-years, endline IDI 7)

This participant understood that girls and women were capable of doing work typically done by men, but that due to gender norms about the acceptability of certain forms of work for women and girls, there may be negative repercussions in the community for those who attempt it. During endline data collection a few adolescent girls also reported professional goals typically reserved for men in Tanzania, such as being a police officer. Moreover, a few adolescent girls were inspired by the character of Nyota to become a motorcycle taxi driver:

*Nyota, when she delivered a baby, her father told her to go to VETA [vocational training], and later she asked her father to buy her a bodaboda so she can start a bodaboda driving business. So, there is no discrimination when it comes to work [. . .] Even us girls we can do anything, not only boys, so it gives me hope to do anything [. . .] I was so inspired.* (Adolescent girl aged 15-years, endline IDI 17)

*I liked [the scene] when Nyota started to drive a bodaboda [. . .] I liked to see a woman drive a bodaboda [. . .] [I dream] to drive a bodaboda [myself].* (Adolescent girl aged 13-years, endline IDI 16)

Like adolescent girls, men and women caregivers also reported adopting more gender equitable attitudes about work after listening to the radio drama, and we did not find any notable differences between the three groups:

*I have listened to the radio drama and realized that adolescent girls are supposed to be protected and be allowed to do the jobs they wish. We should not say that these are women or*

*men jobs, a woman should do any job she wants. I have learned that a woman can do everything.* (Man caregiver aged 46-years, endline IDI 47)

*I learnt that in doing work, there is no man's or woman's work. Like the bodaboda driving, I thought it is for men only, but it is also possible for women to do it [. . .] The girl [Nyota] who went to study how to drive a bodaboda, she changed my thinking. . . Even women, we can do anything.* (Woman caregiver aged 50-years, endline IDI 88)

Furthermore, one man reported learning from the household discussion sessions that sending adolescent boys to school should not be prioritized over adolescent girls:

*We should not leave girls behind [. . .] we should not discriminate [against] them. [. . .] they [household discussion session leaders] were teaching us, for example [that if] there are students and one is a girl student, but [the] boy student is given priority, thinking that a girl can get pregnant [. . .] it should not be that way.* (Man caregiver aged 54-years, endline IDI 58)

These quotes not only demonstrate more gender equitable attitudes about work and education, but highlight how both the adolescent and adult participants were able to critically reflect upon the content of the radio drama, drawing conclusions that went beyond the storylines presented to them and interpreting them within the context of their own lives.

**3.3.2 Women's motivation to engage in income generating activities.** Lastly, we also found that after listening to the radio drama many women viewed starting a small business as the best opportunity to improve their lives, help provide for their families and protect girls from age-disparate transactional sex. These women were inspired by the character of Mama Prita, an older woman in the community who works in the fish market and advises Amali and her sister to be careful around men. When asked the impacts of the radio drama, for example, one woman responded:

*The woman who engaged in different economic activities to foster family economy [Mama Prita], it inspires me to do the same. Sometimes you can give money to your daughter if their father does not have money at the time. If you as woman you're able, then the family will be strong.* (Woman caregiver aged 49-years, endline IDI 90)

By suggesting that she would like to give money to her daughter, this participant refers to poverty as a driver of adolescent girls' engagement in age-disparate transactional sex and explains how she could support her daughter in avoiding these relationships. Similarly, when asked whether the radio drama has impacted their future goals, other women responded:

*Mama Prita was doing her business very well, and also, she takes care of her family well. I would like to be like her. Us as women should follow business things so we can be better too.* (Woman caregiver aged 37-years, midline IDI 87)

*Mama Prita was hardworking. Every day she was hardworking. I need to change that, not to sit as a mother and wait [. . .] I need to be an entrepreneur so I can increase the family income.* (Woman caregiver aged 46-years, endline IDI 83)

These findings suggest that the radio drama catalysed a strong challenge to gendered societal expectations, such as men being the sole providers in the household. It also challenged the expectation of girls providing for themselves once they reach a certain age or leave school, an expectation that puts them at risk of engaging in age-disparate transactional sex.

## 4 Discussion

The findings of this study highlight the potential of the *LINEA* radio drama to promote educational aspirations and gender equitable attitudes about work among adolescents and their caregivers in Tanzania. Given the low rates of secondary school completion among Tanzanian adolescent girls [21, 22] and the high prevalence of gender inequitable norms and sexual violence in the study region [47], these outcomes are particularly pertinent. Past literature has demonstrated that educational attainment is a protective factor for age-disparate transactional sex [18], and evidence from this study suggests that the *LINEA* radio drama has the potential to not only prevent age-disparate transactional sex directly [36], but to also impact important, related contextual factors that drive it: 1) adolescent girls' focus on long-term educational goals and 2) adolescent girls and caregivers increased gender-equitable attitudes about work. Thus, by preventing age-disparate transactional sex, which is a key driver of HIV and unintended pregnancy, the *LINEA* radio drama holds promise in contributing to decreasing the morbidity and mortality of adolescent girls in Tanzania [5, 7].

Despite education already being highly encouraged in the intervention community, we found that the radio drama gave adolescent girls the motivation to persevere, especially when they experienced setbacks. Previous research suggests that when adolescent girls and young women focus on long-term aspirations, such as educational attainment, it can influence them to make safer sexual decisions [27]. Thus, interventions such as the *LINEA* radio drama can leverage these aspirations to deter engagement in age-disparate transactional sex, and guide them towards making sexual decisions that align with their evolving capacities [54]. Although pregnant students are now legally able to remain in school in Tanzania [26], delaying first sexual experiences is still essential, as the responsibilities of caring for a child can be a significant barrier for pregnant adolescent girls and adolescent mothers to continue their education [50]. Stigma and discrimination at the institutional and individual levels in schools also present significant barriers for adolescent girls who wish to continue with their education despite being pregnant or having a child [55, 56]. Moreover, delayed first sex is expected to contribute to decreased HIV, as older adolescents are more likely to negotiate condom use due to more equitable power dynamics vis a vis their sexual partners, and better knowledge about HIV [57]. Therefore, findings from this exploratory study suggest that the *LINEA* radio drama could also contribute towards a set of initiatives to prevent HIV infections among this particularly affected group.

Our study also revealed evidence of more gender-equitable attitudes about work among adolescent girls, and women and men caregivers, creating an enabling environment for changes in gender norms linked to education and employment. The radio drama helped participants accept women working in traditionally male-dominated professions, and motivated women to start small businesses to support their daughters, indicating a departure from traditional male provider norms, and the expectation that adolescent girls provide for themselves once they reach a certain age or leave school. These findings suggest that marketing strategies that encourage women to enrol in traditionally male-dominated vocational trainings could be effective in increasing women's participation in these sectors, thereby contributing to the shift towards more equitable gender norms. Gender equitable norms have been associated with increased empowerment and improved health outcomes for both women and men [58]. Furthermore, evidence from a cash transfer programme evaluation in Tanzania suggests that improved economic circumstances lead to decreased engagement in transactional sex for survival [59]. However, the effectiveness of cash-transfers in preventing such relationships in the general population remains inconclusive [e.g. 60], suggesting that the *LINEA* radio drama could be an important complement to interventions that addresses structural factors such as

poverty in driving age-disparate transactional sex. Findings from this study also highlight the potential of the *LINEA* radio drama to have broader impacts beyond preventing age-disparate transactional sex, by challenging attitudes that promote sexual and gender-based violence more generally and fostering more gender-equitable communities.

While this exploratory evaluation suggests that the *LINEA* radio drama shows promise, more rigorous evaluation is needed to determine causal impact, as well as whether our findings from IDIs with members of the poorest households and who have a family member living with a disability in Shinyanga region, North-western Tanzania are transferable to other populations and regions across Tanzania. The results from this study suggest that the cRCT evaluating *LINEA* planned from 2023–2026 in Mwanza region should also incorporate educational aspirations and gender equitable attitudes about work as secondary outcomes, and should elucidate if changes reported in motivation and attitudes in this study are consistent with changes in social norms and reported behaviours, as evidence suggest the latter are harder to change [42, 43].

## 4.1 Strengths and limitations

This study evaluates an edutainment intervention that uniquely addresses age-disparate transactional sex as a primary outcome. Although this is an exploratory, qualitative study it includes a relatively large sample of diverse ages, and strong themes were generated from the data. A sub-set of participants were interviewed at midline and endline, but we did not find evidence of any meaningful differences between the two timepoints and therefore did not analyse the data longitudinally. This suggests that even after a very short implementation period of three and a half weeks (at midline) respondents reported indications of change, and these persisted through the end of implementation. The lack of difference between the two timepoints may have been because many participants had listened to the entire radio drama by midline. Thus, midline data collection many be more suited for monitoring and evaluation purposes than evaluating intervention impact, helping participant's access interventions if there are any issues, and contributing to flexible and adaptive programming [61].

Despite encouraging participant reports of more gender-equitable attitudes towards work, this evidence was largely driven by the radio drama storyline in which a young woman becomes a motorcycle taxi driver. Thus, further research is needed to determine whether this progressive attitudinal shift extends to other male-dominated forms of employment, as well as whether it leads to behaviour change, as entering a new profession requires significant skill and support, which goes beyond the remit of the *LINEA* radio drama. Finally, we were particularly interested in exploring any potential negative consequences of the intervention, and we did so through specific questions during the interviews and by focusing on this while coding the transcripts. No evidence of negative perceptions of the radio drama were identified, and the limited negative reported impacts we found are thoroughly reported in a forthcoming process evaluation paper of the study. While an experimental impact evaluation is necessary to determine the intervention's effectiveness and long-term sustainability, the preliminary findings from this qualitative evaluation are promising, especially considering the relatively short implementation period of seven weeks.

## 5 Conclusions

This study makes a unique contribution to the literature by providing evidence on the impacts of an edutainment age-disparate transactional sex intervention on educational aspirations, and gender-equitable attitudes about work in Tanzania. This evidence provides insight into how future sexual and reproductive health interventions can be optimized to target these important

outcomes, which are closely related to HIV, morbidity and mortality for adolescent girls in sub-Saharan Africa [9, 14, 15, 18]. This study also highlights the potential wide-reaching impacts of the *LINEA* radio drama to prevent age-disparate transactional sex in particular. If proven effective in the forthcoming cRCT, the *LINEA* radio drama has the potential to be easily and cost-effectively scaled up to increase adolescent girl's motivation to pursue their education and create more gender-equitable communities across Tanzania.

## Acknowledgments

We would like to thank the adolescent girls and their caregivers for their support and participation in this study. We are also grateful to the fieldwork teams from TCRS and AGO who implemented the intervention and conducted the interviews.

## Author Contributions

**Conceptualization:** Marjorie Pichon, Ana Maria Buller, Lottie Howard-Merrill.

**Data curation:** Marjorie Pichon.

**Formal analysis:** Marjorie Pichon, Ana Maria Buller, Yandé Thiaw, Lottie Howard-Merrill.

**Funding acquisition:** Ana Maria Buller, Lottie Howard-Merrill.

**Investigation:** Veronicah Gimunta, Oscar Rutenge, Revocatus Sono.

**Methodology:** Marjorie Pichon, Ana Maria Buller, Lottie Howard-Merrill.

**Project administration:** Marjorie Pichon, Ana Maria Buller, Veronicah Gimunta, Oscar Rutenge, Revocatus Sono, Lottie Howard-Merrill.

**Resources:** Veronicah Gimunta, Oscar Rutenge, Revocatus Sono.

**Supervision:** Marjorie Pichon, Ana Maria Buller, Veronicah Gimunta, Oscar Rutenge, Revocatus Sono, Lottie Howard-Merrill.

**Validation:** Marjorie Pichon, Ana Maria Buller, Veronicah Gimunta, Oscar Rutenge, Revocatus Sono, Lottie Howard-Merrill.

**Writing – original draft:** Marjorie Pichon, Ana Maria Buller, Lottie Howard-Merrill.

**Writing – review & editing:** Marjorie Pichon, Ana Maria Buller, Veronicah Gimunta, Oscar Rutenge, Yandé Thiaw, Revocatus Sono, Lottie Howard-Merrill.

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
