## [Decision Letter · Decision Letter 0]

1 Dec 2023

PGPH-D-23-01790

Qualitative evaluation of an edutainment intervention to prevent age-disparate transactional sex in Tanzania: Changes in educational aspirations and gender equitable attitudes towards work

Dear Dr. Pichon,

Thank you for submitting your manuscript to PLOS Global Public Health. After careful consideration, we feel that it has merit but does not fully meet PLOS Global Public Health’s publication criteria as it currently stands. Therefore, we invite you to submit a revised version of the manuscript that addresses the points raised during the review process.

Please note that we have only been able to secure a single reviewer to assess your manuscript. We are issuing a decision on your manuscript at this point to prevent further delays in the evaluation of your manuscript. Please be aware that the editor who handles your revised manuscript might find it necessary to invite additional reviewers to assess this work once the revised manuscript is submitted. However, we will aim to proceed on the basis of this single review if possible.

We look forward to receiving your revised manuscript.

Kind regards,

Jianhong Zhou

Staff Editor

Journal Requirements:

Additional Editor Comments (if provided):

Reviewers' comments:

Reviewer's Responses to Questions

**Comments to the Author**

1. Does this manuscript meet PLOS Global Public Health’s publication criteria? Is the manuscript technically sound, and do the data support the conclusions? The manuscript must describe methodologically and ethically rigorous research with conclusions that are appropriately drawn based on the data presented.

Reviewer #1: Yes

2. Has the statistical analysis been performed appropriately and rigorously?

Reviewer #1: N/A

3. Have the authors made all data underlying the findings in their manuscript fully available (please refer to the Data Availability Statement at the start of the manuscript PDF file)?

Reviewer #1: Yes

4. Is the manuscript presented in an intelligible fashion and written in standard English?

Reviewer #1: Yes

5. Review Comments to the Author

Reviewer #1: Thank you for the opportunity to review this interesting manuscript on the LINEA intervention. Some considerations for revisions are detailed below:

Minor comments:

1. You use the abbreviation ADTS in the abstract only - please write this out in full.

2. Please briefly expand on the social science theory that informs edutainment in line 127.

3. On line 239 you refer to 20% transcripts - please add the actual number of transcripts dual coded. Were they coded in entirety or for particular themes?

4. Th authors indicate that of the 81 participants recruited, 59 (73%) partook in endline IDIs - what happened to the rest - lost to follow up, any reasons, e.g., relocation

5. Lines 250-260 can be simplified.

Major comments:

1. The manuscript comprises two broad categories - this requires some further refinement into a few more subthemes (subheadings) to improve the readability, flow and coherence.

2. Were there any negative perceptions or negative reported impacts of the intervention that were identified in transcripts? This feels like an important but missing piece especially as the RCT is rolled out.

3. Were there any notable differences in perceptions across genders/age-groups?

6. PLOS authors have the option to publish the peer review history of their article (what does this mean?). If published, this will include your full peer review and any attached files.

**Do you want your identity to be public for this peer review?** For information about this choice, including consent withdrawal, please see our Privacy Policy.

Reviewer #1: No

---

## [Decision Letter · Decision Letter 1]

14 Mar 2024

Qualitative evaluation of an edutainment intervention to prevent age-disparate transactional sex in Tanzania: Changes in educational aspirations and gender equitable attitudes towards work

PGPH-D-23-01790R1

Dear Ms Pichon,

We are pleased to inform you that your manuscript 'Qualitative evaluation of an edutainment intervention to prevent age-disparate transactional sex in Tanzania: Changes in educational aspirations and gender equitable attitudes towards work' has been provisionally accepted for publication in PLOS Global Public Health.

Best regards,

Jhumka Gupta

Academic Editor

Reviewer Comments (if any, and for reference):

Reviewer's Responses to Questions

**Comments to the Author**

1. If the authors have adequately addressed your comments raised in a previous round of review and you feel that this manuscript is now acceptable for publication, you may indicate that here to bypass the “Comments to the Author” section, enter your conflict of interest statement in the “Confidential to Editor” section, and submit your "Accept" recommendation.

Reviewer #2: All comments have been addressed

2. Does this manuscript meet PLOS Global Public Health’s publication criteria? Is the manuscript technically sound, and do the data support the conclusions? The manuscript must describe methodologically and ethically rigorous research with conclusions that are appropriately drawn based on the data presented.

Reviewer #2: Yes

3. Has the statistical analysis been performed appropriately and rigorously?

Reviewer #2: N/A

4. Have the authors made all data underlying the findings in their manuscript fully available (please refer to the Data Availability Statement at the start of the manuscript PDF file)?

Reviewer #2: Yes

5. Is the manuscript presented in an intelligible fashion and written in standard English?

Reviewer #2: Yes

6. Review Comments to the Author

Reviewer #2: The authors did an excellent job addressing prior comments.

Addressing a few minor concerns could further strengthen this manuscript:

1) Typo in line 89: enrol

2) Line 358, participant’s quote: who is the participant referring to in the phrase “they will wonder”. Is “they” the community?

3) Discussion, line 456-458. What about recommending that VETA and other vocational training in Tanzania market male-dominated vocations to girls and women?

4) Limitations section: Are there any limitations to the sampling approach of selecting extremely poor households with a disabled member? Were any of the adolescent girls disabled? If so, how might that have affected the findings?

7. PLOS authors have the option to publish the peer review history of their article (what does this mean?). If published, this will include your full peer review and any attached files.

**Do you want your identity to be public for this peer review?** For information about this choice, including consent withdrawal, please see our Privacy Policy.

Reviewer #2: **Yes: **Thespina (Nina) Yamanis
